# The Spectrum of the Heterozygous Effect in Biallelic Mendelian Diseases—The Symptomatic Heterozygote Issue

**DOI:** 10.3390/genes14081562

**Published:** 2023-07-31

**Authors:** Kateryna Kalyta, Weronika Stelmaszczyk, Dominika Szczęśniak, Lidia Kotuła, Paula Dobosz, Magdalena Mroczek

**Affiliations:** 1School of Life Sciences, FHNW—University of Applied Sciences, 4132 Muttenz, Switzerland; kalitakata05@gmail.com; 2School of Cellular and Molecular Medicine, University of Bristol, Bristol BS8 1TD, UK; an20503@bristol.ac.uk; 3Institute of Psychiatry and Neurology in Warsaw, Genetics Department, 02-957 Warsaw, Poland; dszczesniak@ipin.edu.pl; 4Department of Genetics, Medical University, 20-080 Lublin, Poland; lidialidka@gmail.com; 5Institute of Genetics and Biotechnology, Faculty of Biology, University of Warsaw, Pawinskiego 5A, 02-106 Warsaw, Poland; paula.dobosz@gmail.com; 6University Hospital Basel, University of Basel, 4031 Basel, Switzerland

**Keywords:** symptomatic carrier, heterozygous effect, symptomatic heterozygotes

## Abstract

Heterozygous carriers of pathogenic/likely pathogenic variants in autosomal recessive disorders seem to be asymptomatic. However, in recent years, an increasing number of case reports have suggested that mild and unspecific symptoms can occur in some heterozygotes, as symptomatic heterozygotes have been identified across different disease types, including neurological, neuromuscular, hematological, and pulmonary diseases. The symptoms are usually milder in heterozygotes than in biallelic variants and occur “later in life”. The status of symptomatic heterozygotes as separate entities is often disputed, and alternative diagnoses are considered. Indeed, often only a thin line exists between dual, dominant, and recessive modes of inheritance and symptomatic heterozygosity. Interestingly, recent population studies have found global disease effects in heterozygous carriers of some genetic variants. What makes the few heterozygotes symptomatic, while the majority show no symptoms? The molecular basis of this phenomenon is still unknown. Possible explanations include undiscovered deep-splicing variants, genetic and environmental modifiers, digenic/oligogenic inheritance, skewed methylation patterns, and mutational burden. Symptomatic heterozygotes are rarely reported in the literature, mainly because most did not undergo the complete diagnostic procedure, so alternative diagnoses could not be conclusively excluded. However, despite the increasing accessibility to high-throughput technologies, there still seems to be a small group of patients with mild symptoms and just one variant of autosomes in biallelic diseases. Here, we present some examples, the current state of knowledge, and possible explanations for this phenomenon, and thus argue against the existing dominant/recessive classification.

## 1. Introduction

As a rule of thumb, heterozygous carriers of variants associated with recessive diseases are asymptomatic. This can be confirmed by large population genetic studies and the asymptomatic status of heterozygous family members in segregation analyses [1,2]. Symptomatic heterozygotes, defined as symptomatic carriers of a recessive autosomal disease, are individuals carrying only one copy of the pathogenic/likely pathogenic variant in biallelic autosomal Mendelian diseases. Considering the available literature, a symptomatic heterozygous status in autosomal diseases is extremely rare and has been based on case reports only, although some large studies have suggested an increased risk for some diseases among heterozygotes [3,4]. Although overall very rare, a heterozygous symptomatic state is quite common in some diseases. Also, individuals who carry just one variant of the disease-causing gene and who display quite a severe phenotype are often not defined as symptomatic carriers per se; rather, they are classified as an example of a dominant inheritance with a milder disease form [5,6]. Indeed, sometimes only a fine line exists between these two conditions. In addition, recent population studies investigating links between genetic variants and quantitative traits have shown a spectrum of subclinical phenotypes associated with heterozygosity in some disease variants [3,4]. A broad range of intermediate subclinical phenotypes has suggested significant heterozygous phenotypic effects in some Mendelian biallelic diseases.

An increasing number of case studies now describe patients with only one variant of biallelic Mendelian diseases who display an intermediate phenotype somewhere on the continuum between affected, symptomatic patients and unaffected individuals. These case reports usually describe either a single individual or a whole family displaying various phenotype degrees while having a heterozygous symptomatic carrier status; however, in some cases larger cohorts have been reported [7,8,9]. The limiting factor is, however, the lack of full molecular analysis in most cases so that symptomatic heterozygosity is a hypothesis, and the most probable explanation is that the “second hit” has been missed. In most cases of symptomatic heterozygotes, symptoms are milder than in the disease state [10]. Often, the genetic status of symptomatic heterozygosity is first identified after re-examining the family member of an affected individual because the symptoms are so mild and unspecific that they are not noticed by the patient. Symptoms are usually located somewhere on the continuum between healthy individuals and patients carrying two variants. Correspondingly, symptomatic heterozygotes for genes related to neuromuscular disorders usually present with myalgias [5,6], mild muscular atrophy [11], and for hematological diseases with jaundice and mild symptoms of anemia [12] or familiar mild ptosis among the individuals with only one variant causative for congenital myasthenic syndrome (CMS), but where also cases of recessive CMS occurred [13]. In some cases, only some abnormalities in the laboratory tests may be detected [14,15,16]. This also suggests that it may be a disease phenotypic spectrum between heterozygous individuals and homozygous affected patients.

Investigating symptomatic heterozygotes can also have practical implications in terms of treatment and protection against some diseases. Symptomatic heterozygotes with Familial Mediterranean Fever (FMF) can benefit in childhood from therapy with colchicine [17]. Also, although described in only a few case reports, treatment with enzyme replacement therapy can be beneficial for heterozygous symptomatic late-onset Pompe disease patients [18], although the possibility that the patients present with a real symptomatic heterozygosity is still in dispute. The literature contains one case of a heterozygous patient with aceruloplasminemia who was successfully treated with oral zinc sulfate [19]. In terms of treatment, an important point to highlight is that some of the heterozygous carriers may be influenced by dietary factors. This may be the case for some deficiencies in galactose metabolism that can lead to presenile cataracts. The development of symptoms has been suggested to depend on the amount of galactose uptake by the heterozygotes, as only a small percentage of them develop symptoms [20].

The goal of this narrative review is to revise the available literature and clinical information on the symptomatic heterozygotes in autosomal biallelic diseases and heterozygous effect on the population level for recessive Mendelian diseases. Further, we delineate possible molecular bases of this phenomenon and current knowledge on this topic. We discuss the existing classification of recessive vs. dominant and how to approach the phenotypic spectrum of heterozygosity in recessive Mendelian variants on the systematic level.

## 2. Disease Spectrum and Phenotypic Variability

Symptomatic carriers of autosomal diseases have been described for a wide range of disorders, including neuromuscular, neurological, hematological, and pulmonary diseases (examples are shown in Table 1). Unfortunately, the current knowledge is still based mostly on case reports, and only the tip of the iceberg is revealed. Also, most of the cases reported as examples of “symptomatic heterozygosity” underwent only Sanger sequencing or whole-exome sequencing analysis (WES), which makes missing of the second hit variant the most probable hypothesis.

### Case Reports and Diagnostic Workup

Reported symptomatic heterozygosity is often based on the facts that cases described as carriers of autosomal diseases show milder phenotype than homozygous ones. One example of symptomatic heterozygosity, based mainly on the fact that disease symptoms were milder, is a platelet function disease related to the variants in the *RASGRP2* gene that encodes calcium and diacylglycerol (DAG)-regulated guanine nucleotide exchange factor I associated with the platelet activation function [21]. Patients with biallelic *RASGRP2* variants have been described as having moderate-to-severe bleeding and normal platelet counts and morphology [22]. In a recently described large family, all the carriers but one had mild-to-moderate bleeding, but almost all had epistaxis of variable severity, and two patients had abnormal bleeding following surgery or dental extraction. No other concurrent diagnoses could explain this phenomenon, although it should be highlighted that only a hemostasis disease panel of 92 genes was performed for this family, so an alternative diagnosis, especially the existence of a second variant, cannot be excluded [21]. Symptomatic heterozygotes are also encountered relatively often in metabolic diseases. For example, in GSD type V (McArdle disease; MIM #232600), a glycogen storage disease, heterozygous individuals carrying just one copy of the *PYGM* gene may develop muscle symptoms that are probably related to the critically low residual level (30–40%) of myophosphorylase activity in muscle [23]. A few dominant families have been reported in which some family members were described as symptomatic heterozygotes and others as exhibiting a recessive inheritance pattern. For example, in one case report, an obligate heterozygous patient’s mother presented with myalgia and weakness after exercise, whereas the patient’s father was reported as clinically unaffected. Interestingly, the muscle phosphorylase activity was much lower in this woman (20%) than in the father (45% of normal) [24]. Another metabolic myopathy due to myoadenylate deaminase deficiency (MIM #615511) is one of the most common muscle diseases in humans, with the prevalence of heterozygous individuals reaching even 10% of the population in some ethnic groups [25]. The disease manifests as muscle weakness, myalgia, and fatigue and shows an autosomal recessive inheritance pattern. However, several symptomatic heterozygotes have been identified [26,27,28]. Nevertheless, no consensus has been reached on how to interpret the heterozygous and homozygous variants, given that most of the heterozygotes appear asymptomatic. Oligosymptomatic individuals are also encountered among carriers of cystinuria type 1 and type 2 (MIM #104614 and MIM #604144). They can form renal stones; therefore, measuring renal cystine excretion in relatives of cystinuric patients is useful for identifying heterozygous individuals at risk [29]. Several case reports have also been published on symptomatic heterozygotes in neurological diseases, such as hereditary aceruloplasminemia [30], cerebral vessel disease [31], and α sarcoglycanopathy [11], although a full diagnostic workup was not applied and the patients underwent Sanger sequencing or WES, so the presence of the second variant remains the most probable hypothesis. On the other hand, some cases reported as symptomatic heterozygosity display a variable degree of clinical symptoms between the heterozygous family members, ranging from the full recessive disease phenotype to the mild symptoms or radiological changes only [5,32].

While usually rare, in some diseases symptomatic heterozygosity occurs in several cases. One example is hereditary hemochromatosis (HH). HH is a recessively inherited disease characterized by excessive absorption of dietary iron and its accumulation in the organs. As a consequence, iron accumulates in different organs, such as the liver, heart, pancreas, joints, etc. [9]. HH is caused by an autosomal biallelic variant in the *HFE* gene and the p.Cys282Tyr (rs1800562) pathogenic variant is frequently inherited in a heterozygous state [33]. The overall carrier frequency in the Caucasian population is high and is approximately 1:16. Still, this variant accounts for a symptomatic phenotype only in a minority of cases—around 1.5–3% of heterozygotes show symptoms [33,34]. Some of the heterozygous individuals for p.Cys282Tyr or p.His63Asp (rs1799945) develop only an increased transferrin in the laboratory testing, while others also develop mild symptoms [33,34]. In FMF, symptomatic heterozygotes have been reported in more than 120 cases [7,8,35,36]. In several studies, an extensive search for the second hit has been performed and suggested that a single P/LP variant may be sufficient to cause symptoms. The symptomatic heterozygotes show typical disease symptoms, such as recurrent fever and abdominal attacks, usually in the milder form [7,8,36].

**Table 1 genes-14-01562-t001:** Examples of cases reported as symptomatic heterozygotes in the literature. Molecular diagnosis is often based on Sanger sequencing, and whole diagnostic algorithm, inclusive RNA, and whole-genome sequencing studies are rarely performed.

Disease	Gene	Number of Individuals	Symptoms	Molecular Analysis	Ref.
Non-syndromic, autosomal recessive platelet function disease	*RASGRP2*	5	Prolonged bleeding	Hematology gene panel (92 genes), Sanger for segregation	[21]
α sarcoglycanopathy	*SGCA*	7	Scapular winging to various degrees, proximal muscle weakness, calf hypertrophy in two cases	Sanger, immunohistochemistry, immunoblotting (immunohistochemistry, immunoblotting did not differ in relation to controls)	[14]
FMF	*MEFV*	94	Fever, incomplete abdominal attack	Sanger sequencing, qRT-PCR, Western blot	[9]
Non-classic congenital adrenal hyperplasia	*CYP21A2*	49	Statistically increased 17-OH level only	Sanger sequencing, RFLP	[15]
Autosomal polycystic kidney disease	*PKHD1*	16	Most of the asymptomatic mutation carriers appeared to have increased medullary echogenicity and polycystic liver disease. Increased medullary echogenicity in 6 (5.5%) and multiple small liver cysts in 10 parents (9%) out of 100 heterozygotes were identified	Sanger sequencing	[37]
Congenital myasthenic syndrome	*COLQ*	2	Congenital ptosis	Sanger sequencing	[13]
Hereditary aceruloplasminemia	*ACP*	1	Extrapyramidal and cerebellar-mediated movement disorder	Sanger sequencing	[19]
CSVD	*HTRA1*	10	Cerebral vessel disease	WES, Sanger, validated in the cell culture	[33]
Hereditary hemochromatosis	*HFE*	6	Iron overload at first genetic testing	Sanger sequencing of the exons, exon-intron boundaries and the 5′ untranslated region of the *HFE, HAMP*, *HJV/HFE2*, *TFR2* and *SLC40A1*, RFLP	[37]

RFLP—Restriction fragment length polymorphism.

## 3. Dual Inheritance Mode vs. Symptomatic Heterozygous

For some diseases, both recessive and dominant inheritance patterns have been reported. These include myotonia congenita (Thomsen and Becker myotonia) [38], collagen 6- and 12-related muscle diseases (Bethlem myopathy and Ullrich congenital muscular dystrophy) [39], and glutaryl-CoA dehydrogenase deficiency [40]. The differences between dominant and recessive inheritance for the same gene and symptomatic heterozygotes may be fluent. However, symptomatic heterozygous inheritance is accepted to have only mild symptoms, whereas dominant and recessive inheritance show nearly the same severity for the affected state.

In some diseases, the phenomenon of dual inheritance mode is already well known, especially for the ion channels, whereas for other diseases it has been reported only recently. For the ion channels, the mechanistic association with functional units has been described. Pathogenic/likely pathogenic (P/LP) variants in the *CLCN1* gene (RefSeq NC_000007) underline two distinct types of myotonia congenita (MC): dominant Thomsen’s disease (MIM #160800; ADMC) and recessive Becker’s disease (MIM #255700, ARMC) [41]. Becker’s patients are clinically more affected, exhibit more severe myotonia, transient weakness and generalized muscular hypertrophy [41]. P/LP variants associated with loss of function (e.g., stop, out of frame, splicing) are usually associated with ARMC, whereas missense variants usually underlie ADMC through the dominant negative mechanism. The CLC-1 chloride channel is composed of two separate pores responsible for either slow (common) or fast gating processes. Many variants have been identified as predominantly or exclusively affecting the common gate, explaining the dual inheritance mode [42,43,44,45]. In the case of P/LP variants in *SCN4A*, there are different disease phenotypes in the dominant and recessive forms of the diseases, representing different molecular mechanisms affecting the sodium channel (Table 2) [46]. The variants associated with a dominant inheritance are usually loss-of-function, but missense variants have also been reported. In calpainopathy and dysferlinopathy, a dual inheritance mode has also been suggested, with most of the variants transmitted recessively and a few transmitted dominantly, both missense and loss-of-function variants [5,6,47,48]. For some diseases, a semidominant inheritance pattern has been well described. An example of a disease inherited in an autosomal semidominant manner is familial hypercholesterolemia (FH) caused by a defect in the *PCSK9*, *LDLR*, and *APOB* genes. If an individual has biallelic (homozygous or compound heterozygous) pathogenic variants in one of these three genes—a condition referred to as homozygous FH (HoFH)—the presentation becomes more severe with earlier onset of features [49].

## 4. Population Studies and Phenotypic Spectrum

Interestingly, large population studies seem to confirm the existence of a spectrum of mild subclinical phenotypes related to disease at the population level. A study from the UK biobank encompassing 487,409 participants showed 102 significant associations, indicating that many disease-associated recessive variants can produce mitigated phenotypes in heterozygous carriers. These include, most notably, associations with a cystic fibrosis *CFTR* p.Phe508del (MAF = 1.6%) variant, that in carriers showed significant associations (q value < 0.05) with asthma (OR = 1.12; 95% CI, 1.06–1.17), aspergillosis (OR = 2.60; 95% CI, 1.63–4.13), bronchiectasis (OR = 1.40; 95% CI, 1.20–1.61), and duodenal ulcer (OR = 1.30; 95% CI, 1.15–1.45) [3]. Further examples include a *POR* missense variant associated with Antley–Bixler syndrome that showed a 1.76 (SE 0.27) cm increase in height and an *ABCA3* missense variant implicated in interstitial lung disease that was associated with reduced forced expiratory volume during the first second (FEV1)/forced vital capacity (FVC) ratio. However, this is not the case for all diseases. The UK study found no association between *SMN1* carrier status (one *SMN1* copy) and three traits related to neuromuscular function, namely walking speed, grip strength, and the FEV1/FVC ratio [3]. Other recent large biobank studies from a Finnish cohort confirmed these results and found a phenotypic effect in the Mendelian diseases, likely pathogenic heterozygous variants in 203 likely disease genes associated with recessive inheritance in a phenome-wide association study (pheWAS). The Finnish population shows a strong founder effect, so the enriched variants were chosen in order to reach a sufficient statistical study power. In particular, a heterozygous effect has been identified for the variants *SERPINA1*, *NPHS1*, *CASP7*, and *GJB2* that are associated with biallelic disease [4]. The authors suggested that these be reclassified as “recessive, with rare expressing heterozygotes” and summarized that the inheritance of many known Mendelian variants cannot be adequately described by a conventional definition of dominant or recessive [4].

## 5. Molecular Background and Possible Influencing Factors

The status of a symptomatic carrier, from a genetic point of view, is still unclear. Possibly, some additional genetic modifiers or mutational burdens contribute to being a symptomatic carrier. Undoubtedly, the population studies on heterozygotes suggest that most heterozygotes for autosomal recessive disorders are asymptomatic. What is the molecular basis for symptomatic heterozygosity, and what other alternative diagnoses can be considered?

### 5.1. Types of Variants

The type of variant also seems important in heterozygous carriers of autosomal disease; however, due to the small number of individuals, no sufficient studies have been performed [53]. A dominant effect may therefore be a result of a single functional allele not producing sufficient gene product for proper function, resulting in haploinsufficiency, while some variants may follow a gain-of-function mechanistic model [53]. A dominant negative (DN) effect means that the expression of a mutant protein interferes with the activity of a wild-type protein. The DN model has been suggested to operate in calpainopathy [5] and dysferlinopathy [6], for example. Here, not only the deleteriousness of the variant but also its structural location plays a role. This is especially the case for ion channels, where variants located across both pore domains can induce disease even in the heterozygous state. The symptoms are usually milder in the heterozygotes than in the homozygotes and the disease starts later in life [5,6]. Interestingly, in pseuedodomiant inheritance, most of the variants also appear to be null mutations [54,55], but missense variants have been described as well [56,57], highlighting the possibility that in symptomatic heterozygous individuals may indeed come to the pseudodominant inheritance.

### 5.2. (Deep) Splice Site Variants and Genetic Modifiers

The most obvious differential diagnostic possibility is that a second pathogenic variant has been missed. Several diseases are known in which the variant has been reported first as dominant, but further analysis revealed a second variant later in the diagnostic process. An example is the pseudo-dominant inheritance reported for titinopathies [58,59]. Usually, only the flanking regions around ±10 bp from the exon are sequenced, and even variants described as pathogenic but slightly further from the exon, such as −20 bp variants, are often missed [60]. Although the general belief was that only splicing variants located near the introns would have a detrimental effect, recent studies have reported deep splicing variants in several cases, such as recently discovered deep splicing variants in *DYSF* [61] and *CAPN3* [62] that cause the exotification of an intron. A second, quite common hypomorphic variant can also be overlooked, especially considering that the populational frequency in some populations may be greater than an agreed population frequency of 1%. This is especially the case for endemic populations where pseudodominant inheritance has been described [17]. For example, the *ABCA4* p. Gly863Ala (rs76157638) variant described as having conflicting interpretations of pathogenicity according to ClinVar that causes a mild form of Stargardt disease has a carrier frequency of 1.8% in Europe [63], the *GJB2* p.35delG (rs80338939) variant causing congenital deafness has a carrier frequency of 2.9% in southern Europe [64], the *ERCC8* p.Tyr322Term (rs121434323) variant causing Cockayne syndrome has a carrier frequency of 6.8% in Israeli Christian Arabs [65], the *SPG7* p.Ala510Val (rs61755320) variant associated with adult-onset neurodegenerative disease has a carrier frequency of 3–4% in the UK population [66], and the *WARS2* p.Trp13Gly (rs139548132) hypomorphic variant associated with a tremor–parkinsonism syndrome occurs in ~0.5% of the general European population [67]. Some authors argue that the late age onset in heterozygotes speaks against the presence of a second variant. However, a later age onset has been described in cases with a second hypomorphic variant [68]. One potential mechanism of the second variant is that it affects splicing by creating multiple splicing isoforms, such as in the case of *CAPN3* [69] and *AMDP1* [70].

Sometimes the role of the second variant cannot be easily classified as a modifier of pathogenic variants; however, population and clinical studies suggest some detrimental effects. Individuals who are compound-heterozygous for p.His63Asp and p.Cys282Try in *HFE* are at an increased risk of iron overload, with an estimated clinical penetrance of 1–2% [34,71], suggesting that a second variant in the same gene may play the role of a genetic modifier. A role of the cis-eQTL in the expression of the deleterious gene copy has been suggested as a potential mechanism of modifying penetrance [72]. According to this mechanism, expression of the deleterious copy should be modified by the cis-eQTL influencing the expression of the functional copy. However, the studies investigating an association test between opposite-haplotype genotypes and mitigated phenotype did not show any correlations, despite being well powered [3].

### 5.3. Oligogenic Inheritance and Mutational Burden

Digenic/oligogenic inheritance has been identified in a number of disorders, including neurodevelopmental disorders, cardiac disorders, eye diseases, and rare multisystemic disorders, such as Bardet–Biedl syndrome [73,74,75]. Moreover, it was shown that some were heterozygous carriers for individual mutations in more than one gene involved in these functionally related pathways. In isolation, heterozygosity for each mutation was clinically irrelevant, but concurrent heterozygosity was synergistic, leading to clinically relevant biochemical derangements. This model of “synergistic heterozygosity” can be very useful for the understanding of complex phenotypes [76,77].

Larger studies suggest a role of complex inheritance traits, including mutational burden, in neuromuscular and peripheral nerve disorders [78]. The impact of genetic modifiers/second genetic variants can also dictate the phenotype. An example of digenic inheritance, where one variant acts as a phenotype modifier, is the variation in *TIA1/SQSTM1*. Identical variants in *SQSTM1* are associated with four different phenotypes: amyotrophic lateral sclerosis (ALS), frontotemporal dementia, Paget’s disease of the bone, and distal myopathy. The *TIA1* (p.Asn357Ser; rs116621885) variant acts as a modifier dictating muscular phenotype when inherited together with a pathogenic *SQSTM1* variant [79]. Patients who display clinical evidence of energy metabolism disorders and exhibit concurrent partial enzymatic deficiencies in several energy-generating pathways have also been reported. Some of these patients were confirmed as heterozygous carriers for individual mutations in more than one gene involved in these functionally related pathways. In isolation, heterozygosity for each mutation was clinically irrelevant, but the concurrent heterozygosity was synergistic, leading to clinically relevant biochemical derangements. This model of “synergistic heterozygosity” can be very useful for understanding complex phenotypes [77]. Some unsolved cases with variants in more than one gene related to metabolic disorders have been reported. These cases were remarkable in that concomitant reductions were identified in more than one enzyme involved in four different energy metabolism pathways, specifically fatty acid β-oxidation, oxidative phosphorylation, glycogenolysis, and high-energy phosphate recycling [77]. In HH, enhanced iron loading and manifestation of HH may occur in persons with heterozygous *HFE* mutations and additional mutations in the hepcidin (*HAMP*), *HJV*, or *TFR2* genes, suggesting the impact of other genetic modifiers for occurrence of symptoms in heterozygous carriers of biallelic diseases [71]. Recently, an oligogenic background has been suggested to explain male infertility, with the accumulation of several rare heterozygous variants in distinct but functionally connected genes considered to be a probable disease mechanism in a mice model [80]. Thus, the coexistence of multiple single-gene variants in an individual appears to have resulted in complex phenotypes characterized by the emergence of symptoms atypical of the heterozygous state of any of the individual underlying disorders.

In fact, in some cases of symptomatic heterozygosity, digenic or polygenic inheritance may take place instead of a “real” symptomatic heterozygosity. Statistical analysis of the unsolved affected groups may provide indirect proof of digenic inheritance. For example, an oligogenic disease mechanism was suggested for Charcot Marie Tooth neuropathy (CMT) based on statistical and functional studies [78].

### 5.4. Epigenetic Factors

Epigenetic factors have long been proven to play a role in X-linked diseases; however, their contribution to autosomal diseases is still being discussed. For approximately 5% of CpG sites, a significant (>30%) difference in DNA methylation can exist between the two alleles. Consequently, methylation of the wild-type allele can lead to symptoms in heterozygous individuals [81,82]. Genetic variation can also affect histone modifications, which can alter chromatin accessibility and result in allele-specific binding of transcription factors. This, in turn, can cause the expression of only the mutated allele in heterozygous individuals. In age-related diseases, including cancers, cardiovascular diseases, and neurodegeneration, the gene quite commonly will be expressed only from one allele. Phenotypic variation in the case of heterozygotes with autosomal diseases can be explained by random allelic expression (RAE) of many autosomal genes.

Several examples have indicated that symptomatic heterozygosity may be linked to methylation and other epigenetic factors. In the case of *RYR1*-related myopathies, the *RYR1* wild-type gene can be epigenetically silenced in heterozygous individuals. Here, a tissue-specific genomic imprinting is observed, in which the skeletal muscle gene is expressed only from the paternal allele. Zhou et al. (2006) showed that the maternal allele of the RYR1 gene is silenced by DNA methylation and that differentially methylated regions (DMRs) are not localized in the CpG islands in the 5′ region of the gene [53]. Moreover, another epigenetic mechanism, histone deacetylation, appears to have no role in this case. The skewed prevalences between the sexes further indicate the possibility of an epigenetic factor impact in several other diseases. A sex dependence for the penetrance of inherited mutations has been reported in a variety of different heritable disorders, including hereditary hemochromatosis (*HFE*) [71], hypertrophic cardiomyopathy (*MYBPC3, MYH7*) [83], arrhythmogenic right ventricular dysplasia/cardiomyopathy (*PKP2*) [84], long QT syndrome (*KCNQ1*, *KCNH2*, and *SCN5A*) [85], and hypokalemic periodic paralysis (*CACNA1S* and *SCN4*) [86,87]. Sex dependence is also described for cerebral autosomal recessive arteriopathy with subcortical infarcts and leukoencephalopathy (CARASIL). Here, males have vascular risk factors more frequently than female CARASIL patients. In CARASIL, more than 50 symptomatic carriers have been reported, although most parents were asymptomatic [88]. For CARASIL, both gender and environmental factors may be involved, as strict risk factor control decreases the risk.

### 5.5. Environmental Factors

The influence of environmental factors can be a further factor triggering symptomatic heterozygosity and has been suggested for a number of diseases. For example, in FMF, other physiological, environmental, or unidentified conditions that could cause inflammation might also affect the protein threshold or the levels of pyrin required to fight the inflammation. The risk of lung disease may be increased in antitrypsin *MZ* heterozygotes depending on their environmental exposure, such as smoking and occupational exposure (including exposure to environmental pollutants used in agriculture, mineral dust, gas, and fumes) [89]. Diet is one of the most important factors regulating gene expression. Thus, an inherited predisposition to obesity, exemplified by the association between dietary fat intake and obesity in carriers of the *PPARG2* p.Pro12Ala (rs1805192) variant, is modifiable by diet [90]. The variants can also influence laboratory test results in heterozygous carriers for hereditary fructose intolerance or homocystinuria and are suggested to cause cataract symptoms in cases of heterozygous galactose metabolism defects [20]. For other diseases, such as those associated with *HFE*, a high degree of variability in expression is observed among p.His63Asp homozygotes and compound heterozygotes, and environmental factors, including alcoholism and viral hepatitis, have been suggested to influence phenotypic expression. Indeed, a recent study by Aguilar-Martinez et al. found no association between iron overload in p.His63Asp homozygosity and the influence of other *HFE*-related mutations [33]. Other studies have confirmed and extended the concept that the heterozygous mutation in isolation is insufficient in itself to cause clinically significant iron overload, and that the presence of an additional modifying factor, such as excessive alcohol consumption, may be necessary for the expression of an overt iron overload [91].

## 6. Limitations

Although reported, the existence of symptomatic heterozygotes and their classification is still discussed. The presence of symptoms in heterozygous patients is often attributed to the inability of screening methods to identify all potentially pathogenic variants. The most widely discussed concept is incomplete diagnostics, with most of the patients undergoing only Sanger sequencing or whole-exome sequencing only. Therefore, re-assessing the accuracy of this widely used method, such as exome and Sanger sequencing, became a necessity. Heterozygous symptomatic patients should be analyzed further through genome sequencing and other omic methods, such as RNA-Seq and proteomics. There are also several other obstacles in analyzing heritable traits across families, such as variable expression or penetrance. Also, the real frequency of symptomatic heterozygous patients is often not known, as very few cases are reported in the literature and scientists may be criticized for reporting not fully diagnosed cases. There is not a clear database of such cases and the diagnostic procedures are also not standardized between the counties and laboratories, so that it is quite impossible to gather a large homogeneous cohort for further studies. For large population studies on the heterozygous effect of the recessively inherited variants, the largest obstacle is the availability of the appropriate cohort. Often, both the detailed phenotypic data are missing and the cohort is too small to reach a significant statistical power or phenomena at the population level, such as a bottleneck effect or a great level of consanguinity in some cohorts [7]. Further, functional studies on the mechanism of the symptomatic heterozygosity phenomenon are often missing and the studies were performed only with Sanger sequencing, not including RNA-Seq and/or WGS. For symptomatic heterozygotes epigenetic regulation analysis has been performed only for a few diseases and often the mechanism is only suspected. 

## 7. Conclusions

Symptomatic heterozygotes remain a controversial entity, with supporting evidence sometimes based only on a few case reports. Today, genetic heterogeneous traits with variable expressivity inheritance models are still being debated. Genetic counseling should consider the complexity of genetic factors; however, how to approach symptomatic heterozygosity systematically is unclear. The group of symptomatic heterozygotes, when compared to the heterozygous at the population level, is very small; thus, testing is not recommended, especially since the factors contributing to symptom occurrence are unknown. Detailed studies elucidating the mechanism of the symptomatic heterozygosity phenomenon are needed. A lack of understanding of the molecular basis of these disorders and their systematic interpretation can impose burdens on treatment. Genetic counseling should consider the complexity of these genetic factors and the limitations of current screening.

## Figures and Tables

**Table 2 genes-14-01562-t002:** Examples of diseases with dual, dominant and recessive, inheritance mode.

Disease	Gene	Phenotype Recessive vs. Dominant	Citation
Miyoshi myopathy (dysferlinopathy)	*DYSF*	Later onset and milder symptoms, inclusive weakness and muscular atrophy; onset in the 4th to 6th decade in the dominant form	[6]
Calpainopathy	*CAPN3*	Late-onset mild proximal weakness in a dominant form vs. onset in the 1st or 2nd life decade manifesting mostly with a proximal weakness in the recessive form. Some individuals with the dominant form have only myalgias or hyperCKemia, some of them asymptomatic even if in their 70s	[5]
Thomson and Becker disease	*CLCN1*	Myotonia congenita, main symptoms are: muscle stiffness, electrophysiological and clinical myotonia, and weakness. Milder symptoms and no muscle atrophy in the dominant form.	[41]
Hyperkalemic periodic paralysis and congenital myopathy	*SCN4A*	Muscle weakness, myotonia, electromyographic myotonia in a dominant inheritance, in a recessive inheritance or myasthenic syndrome by hypomorphic variants	[46]
RYR1-related myopathy	*RYR1*	Different phenotypes: Dominant P/LP variants have been associated with central core disease and/or a susceptibility to malignant hyperthermia. Recessive mutations usually associated with multiminicore disease, centronuclear myopathy, and congenital fiber type disproportion	[50,51,52]
Becker and Ulrich myopathy	*COL6* *COL12A1*	More severe disease form in homozygosity	[39]

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
