# Peer review of "The Spectrum of the Heterozygous Effect in Biallelic Mendelian Diseases—The Symptomatic Heterozygote Issue"

_genes, 2023, doi:10.3390/genes14081562_

Round 1

Reviewer 1 Report

The authors in the manuscript describe the so-called symptomatic heterozygotes. These are carriers of an autosomal recessive allele who display symptoms of the specific disorder. This is a very interesting topic as there has been a debate among geneticists in recent years about the concept of dominant and recessive disorders. 
The paper contains all necessary parts and is easy to read, since the language from one side is simple but from other hand is scientifically acceptable.   
 I believe that it is an interesting paper, but it requires some editing.
First of all, you cite many different case reports as examples of symptomatic heterozygosity. Could you give more information about the molecular-genetic analyses performed in these cases? Was RNA- sequencing or proteomics analysis done, was whole genome sequencing done? If not, the simple answer is that the second mutation was not found and these are not real examples of symptomatic heterozygosity. 
Page 4, lines 193-195 - Your sentence about the Finnish cohort is not clear. Please, elaborate it. 
Page 6, lines 238-239 - It seems like something from this sentence is missing, it is not complete - However, the calpain expression in the muscle biopsies that revealed 5-15% of the resting calpain.
Page 6, lines 263 - 265 - If there was another mutation in the RYR1, than this is not an example of symptomatic heterozygosity. 
Page 9, line 336,  but a for which a second variant has subsequently been identified - there seems to be a mistake in the sentence, please, correct it. 
Page 9, line 344 - 351 - the second variant will be overlooked, if the analysis involved only panel testing, for example, so it depends on the type of analysis, not on the carrier prevalence. I disagree with your statement.

Author Response

Dear Reviewer,

Thank you for inviting us to submit a revised version of our manuscript "The spectrum of the heterozygous effect in biallelic Mendelian diseases – the symptomatic heterozygotes"

We introduced major changes to the manuscript according to the reviewers’ suggestions and shortened the manuscript significantly. 

If any questions arise please do not hesitate to contact us.

Thank you for your time and comments which helped us to improve the manuscript substantially.

Comments and Suggestions for Authors

First of all, you cite many different case reports as examples of symptomatic heterozygosity. Could you give more
information about the molecular-genetic analyses performed in these cases? Was RNA- sequencing or
proteomics analysis done, was whole genome sequencing done? If not, the simple answer is that the second mutation was not found and these are not real examples of symptomatic heterozygosity.

We fully agree with the Reviewer and added information about the molecular analysis in Table 1. We also added a comment stating that although these examples were reported as symptomatic heterozygosity in the literature, given the performed analysis, the most probable hypothesis is the presence of the "second hit".

Page 4, lines 193-195 - Your sentence about the Finnish cohort is not clear. Please, elaborate it.

We elaborated it as follows (Line 226-235): "Other recent large biobank studies from a Finnish cohort confirmed these results and found a phenotypic effect in the Mendelian diseases likely pathogenic heterozygous variants in 203 likely disease genes associated with recessive inheritance in a phenome-wide association study (pheWAS). The Finnish population shows a strong founder effect so the enriched variants were chosen in order to reach a sufficient statistical study power. That. In particular, a heterozygous effect has been identified for variants SERPINA1, NPHS1, CASP7, and GJB2 that are associated with biallelic disease [4]. The authors suggested that these be reclassified as “recessive, with rare expressing heterozygotes” and summarized that the inheritance of many known Mendelian variants cannot be adequately described by a conventional definition of dominant or recessive [4]".

Page 6, lines 238-239 - It seems like something from this sentence is missing, it is not complete - However, the calpain expression in the muscle biopsies that revealed 5-15% of the resting calpain.

This sentence has been removed in the revision course.

Page 6, lines 263 - 265 - If there was another mutation in the RYR1, than this is not an example of symptomatic
heterozygosity.

We fully agree with the Reviewer. This sentence has been removed in the revision course.

Page 9, line 336, but a for which a second variant has subsequently been identified - there seems to be a mistake in the sentence, please, correct it.

We corrected it to: "Several diseases are known in which the variant has been reported first as dominant. However, further analysis revealed a second variant later in the diagnostic process".

Page 9, line 344 - 351 - the second variant will be overlooked, if the analysis involved only panel testing, for example, so it depends on the type of analysis, not on the carrier prevalence. I disagree with your statement.

We agree with the reviewer and changed it to: "A second, quite common hypomorphic variant can also be overlooked, especially considering that the populational frequency in some populations may be greater than an agreed population frequency of 1% and given that often the filter frequency of 1% is considered a cutoff for genetic data analysis". 

This clarifies that the second variant is often missed because of the filtering criteria applied, not because of the frequency per se.

Reviewer 2 Report

The manuscript by Kalyta et al entitled “The spectrum of the heterozygous effect in biallelic Mendelian diseases – the symptomatic heterozygotes issue” focus a controversial and yet poorly investigated topic of the association to disease in carriers of a single pathogenic variant. Overall, this review paper is well written and its addresses an interesting problem. However, the manuscript as it is, seems to be too long, to contain several inaccuracies (see comments bellow) and the provided examples are kind of repeated throughout the text. Could the authors be more concise when reporting the examples of HFE and CFTR heterozygous states? This review could benefit from a reorganization where HFE and CFTR were presented separately in two gene specific sections.        

Other comments:

1-     Line 23: consider the use of “ …often detected later in life.

2-     Line 32: concerning the “somatic mosaicism” it is not clearly addressed in the manuscript.

3-     The authors should shorten the general introduction by removing specific examples such as the ones of HFE, CFTR and CYP21A genes. Those could be later described in the manuscript under a more appropriated heading.  

4-     Line 72-81: This paragraph should be reduced or removed. The authors could simply state that their work is center in autosomal disorders. Consistently, remove as well lines 312-330 about X-linked pathogenic variants.

5-     The authors must use the same variant nomenclature along the manuscript. Protein variants should be preceded by a p. such as p.Cys282Try and reference SNP ID (rs) should be also included. There is no need to use single letter abbreviations such as C282Y.

6-     Line 113: This review contains an exhaustive or non-exhaustive revision of the literature concerning symptomatic heterozygous? Please clarify this issue.

7-     Line 128-130: The sentence is confusing; the authors should rephrase it. For example: those. “Those are known to occur in quite common diseases, such as hemochromatosis and mediterranean fever caused by variants in the HFE and (?) genes, respectively”. The authors could consider the inclusion of the introductory HFE description (line 58- 66) here.

8-     Line 144: It should be symptomatic heterozygous instead of heterozygous carriers. Heterozygous (or variant) carriers can be asymptomatic or symptomatic.

9-     Line 149: correct the sentence to: “Another metabolic myopathy due to myoadenylate deaminase deficiency (MIM #615511) …”

10-  The authors should check carefully Table 1, for wrong text alignments and extra numbers (e.g. 1HFE). For PKHD1 gene, the description shown in the “Number of individuals” column should be moved to the “Symptoms” one. The authors must pay attention that only symptomatic heterozygous should be considered.  

11-  Line 225-234: The description of variants affecting CIC-1 could be simplified. The authors should avoid the details on how the 3D structure is affected, particularly when no figure is provided.

12-  Lines 244 and 247: Is HF or FH? Please correct the wrong abbreviation.

13-  Please check Table2: a) Provide meaning of LGMD2A and LGMR2A. b) Complete the description of CLCN1 phenotype. c) No information is given for SCN4A disease and phenotype.

14-  Line 270: If heterozygous do not show symptoms they cannot be designated as symptomatic. Please correct the sentence accordingly.

15-  Line 276-284: Sickle cell anemia is caused by a specific variant of HBB gene, where the pathogenic allele used to be named as “S” and the wildtype as “A”. Please correct accordingly and label variants with updated nomenclature (HGVS) employed also by genomic databases of human sequence variation.

16-  Line 285- 287:  There is no proven selective advantage of CFTR heterozygosity, indeed some studies claim that the frequency of del508 mutation might have been reached in the absence of selection. In other words, those are only hypothesized positive phenotypes of CFTR mutation carriers.

17-  Line 287 – 301: These sentences should be removed given that authors describe several associations with disease, which do not fit the topic of an evolutionary advantage. Also, there is no proof of such phenotype have been selected against (purifying selection).

18-  Line 323 – 326: These sentences are misleading a gain-of-function cannot be caused by a null mutation. Please rephrase it.            

19-  Lines 365-367: Why is the haemostatic potential of the individual compromised? Please clarify the sentence.

20-  Lines 398- 400: What the authors mean by multisystem proteinopathy modifying the phenotype to the one comparable with a different disease? The sentence is confusing.

21-  Line 473- 474: Similar to the 15th comment made above for the Sickle cell anemia, the authors must identify the affected gene and the meaning of MZ genotype according to updated genetic variant nomenclature. Briefly, the risk of lung disease may be increased in moderate alpha-1-antitrypsin deficiency caused by a combination of the pathogenic variant of SERPINA1 named as Z with the wildtype M.

22-  Line 483- 499: If the authors are refereeing to epigenetic factors this paragraph should be moved to the former section (5.4)?

23-  The text of the manuscript is self-explanatory; thus Figure 1 does not seem necessary and can be removed.

24-  Line 527: The authors should explain how symptomatic heterozygous were confirmed by western blot staining, or remove the sentence.

Author Response

Dear Reviewer,
Thank you for inviting us to submit a revised version of our manuscript "The spectrum of the heterozygous effect in biallelic Mendelian diseases – the symptomatic heterozygotes issue"
Please find in the following our detailed responses to each point raised during the review process. We introduced changes to the manuscript according to the reviewers’ suggestions, shortened the manuscript and changed its structure.

We would like to thank you for your time and comments which helped us to improve the manuscript substantially.

Comments:

However, the manuscript as it is, seems to be too long, to contain several inaccuracies (see comments bellow) and the provided examples are kind
of repeated throughout the text.

We shortened the manuscript and deleted the repeated examples.

Could the authors be more concise when reporting the examples of
HFE and CFTR heterozygous states? This review could benefit from a reorganization where HFE and CFTR were presented separately in two gene specific sections.

We reorganized the manuscript, provided a specific HFE section and decided to delete CFTR section.  

Other comments:
1- Line 23: consider the use of “ …often detected later in life.
We corrected it accordingly.

2- Line 32: concerning the “somatic mosaicism” it is not clearly addressed in the manuscript.
We deleted "somatic mosaicism"

3- The authors should shorten the general introduction by removing specific examples such as the ones of HFE, CFTR and CYP21A genes. Those could be later described in the manuscript under a more appropriated heading.

We deleted the specific examples from the introduction and described them in the manuscript body text or in Table 1.

4- Line 72-81: This paragraph should be reduced or removed. The authors could simply state that their work is center in autosomal disorders. Consistently, remove as well lines 312-330 about X-linked pathogenic variants.
We removed both paragraphs.

5- The authors must use the same variant nomenclature along the manuscript. Protein variants should be preceded by a p. such as p.Cys282Try and reference SNP ID (rs) should be also included. There is no need to use single letter abbreviations such as C282Y.

We unified the nomenclature accordingly. SNP ID was provided when the variant was mentioned for the first time.

6- Line 113: This review contains an exhaustive or non-exhaustive revision of the literature concerning symptomatic heterozygous? Please clarify this issue.
We removed it in the review course.

7- Line 128-130: The sentence is confusing; the authors should rephrase it. For example: “Those are known to occur in quite common diseases, such as hemochromatosis and mediterranean fever caused by variants in the HFE and (?) genes, respectively”. The authors could consider the inclusion of the introductory HFE description (line 58- 66) here.

Original lines 128-130 have been removed in the revision course and we added a special paragraph dedicated to HFE with a disease description.

8- Line 144: It should be symptomatic heterozygous instead of heterozygous carriers.Heterozygous (or variant) carriers can be asymptomatic or symptomatic.

We agree with the reviewer and changed it accordingly.

9- Line 149: correct the sentence to: “Another metabolic myopathy due to myoadenylate deaminase deficiency (MIM #615511) …”
We corrected it accordingly.

10- The authors should check carefully Table 1, for wrong text alignments and extra numbers
(e.g. 1HFE). For PKHD1 gene, the description shown in the “Number of individuals” column should be moved to the “Symptoms” one. The authors must pay attention that only symptomatic heterozygous should be considered.
We edited Table 1 extensively.

11- Line 225-234: The description of variants affecting CIC-1 could be simplified. The authors should avoid the details on how the 3D structure is affected, particularly when no figure is provided.
We delated this part.

12- Lines 244 and 247: Is HF or FH? Please correct the wrong abbreviation.

We corrected it to FH

13- Please check Table2:

a) Provide meaning of LGMD2A and LGMR2A.

We changed it to calpainopathy

b) Complete the description of CLCN1 phenotype.

We completed the description

c) No information is given for SCN4A disease and
phenotype.
We added this information and edited Table 2 extensively.

14- Line 270: If heterozygous do not show symptoms they cannot be designated as symptomatic. Please correct the sentence accordingly.
This part has been deleted in the revision course.

15- Line 276-284: Sickle cell anemia is caused by a specific variant of HBB gene, where the pathogenic allele used to be named as “S” and the wildtype as “A”. Please correct accordingly and label variants with updated nomenclature (HGVS) employed also by genomic databases of human sequence variation.
This part has been deleted in the revision course.

16- Line 285- 287: There is no proven selective advantage of CFTR heterozygosity, indeed some studies claim that the frequency of del508 mutation might have been reached in the absence of selection. In other words, those are only hypothesized positive phenotypes of CFTR mutation carriers.

We agree with the Reviewer and delated this part.

17- Line 287 – 301: These sentences should be removed given that authors describe several associations with disease, which do not fit the topic of an evolutionary advantage. Also, there is no proof of such phenotype have been selected against (purifying selection).

We removed this part.

18- Line 323 – 326: These sentences are misleading a gain-of-function cannot be caused by a null mutation. Please rephrase it.
It deleted this sentences in the revision course.

19- Lines 365-367: Why is the haemostatic potential of the individual compromised? Please clarify the sentence.
We decided to shorten the manuscript and reduce the number of examples. This part has been deleted in the review course.

20- Lines 398- 400: What the authors mean by multisystem proteinopathy modifying the phenotype to the one comparable with a different disease? The sentence is confusing.

We edited this sentence to:  "An example of digenic inheritance, where one variant acts as a phenotype modifier, is the variation in TIA1/SQSTM1. Identical variants in SQSTM1 are associated with four different phenotypes: amyotrophic lateral sclerosis (ALS), frontotemporal dementia, Paget’s disease of the bone, and distal myopathy. The TIA1 (p.Asn357Ser; rs116621885) variant acts as a modifier dictating muscular phenotype when inherited together with a pathogenic SQSTM1 variant [81]."  

21- Line 473- 474: Similar to the 15th comment made above for the Sickle cell anemia, the authors must identify the affected gene and the meaning of MZ genotype according to updated genetic variant nomenclature. 

This part has been deleted in the revision course.

22- Line 483- 499: If the authors are refereeing to epigenetic factors this paragraph should be moved to the former section (5.4)?

The part regarding CARASIL, where we referred to the epigenetic factors was moved to paragraph 5.4

23- The text of the manuscript is self-explanatory; thus Figure 1 does not seem necessary and can be removed.
We agree with the Reviewer and removed Figure 1

24- Line 527: The authors should explain how symptomatic heterozygous were confirmed by western blot staining, or remove the sentence.
We removed the sentence.